# Integrated Use of Local and Technical Soil Quality Indicators and Participatory Techniques to Select Them. A Review of Bibliography and Analysis of Research Strategies and Outcomes

**Greta Braidotti, Maria De Nobili and Lucia Piani ***

Agricultural-Food, Environment and Animal Care Department, University of Udine, 33100 Udine, Italy; greta.braidotti@uniud.it (G.B.); maria.denobili@uniud.it (M.D.N.)
* Correspondence: lucia.piani@uniud.it

**Abstract:** Climate change has strong impacts on soil conservation and agricultural productivity, with severe consequences on smallholders in developing countries, but virtually no research has been carried out so far on this issue. Therefore, it is necessary to foster the implementation of participatory projects to help communities deal with new difficulties. Sustainable soil management can reduce and even reverse land degradation, helping farmers to adapt to climate change effects. Progress toward sustainability cannot be implemented in small rural communities regardless of local knowledge, which can be addressed using participatory techniques. To this purpose the choice and use of indicators is essential to carry out correct assessments of soil vulnerability integrating local and technical knowledge. The purpose of this review was to study how the problem of building a set of integrated indicators to assess soil quality has been addressed so far and which participatory techniques have been more successfully employed, analyzing studies carried out in rural communities of developing countries. We found out that there is a lack of participated studies dealing with environmental issues. Those that do so address them only indirectly, being centered on present agricultural problems. The studies rarely feature a collaboration with social science experts, consequently the use of participatory techniques lacks protocols and a standardized nomenclature to help in the transfer and generalization of experiences. Women are rarely involved and nearly exclusively in African countries: this could be related to social and cultural conditions, but needs more attention. Different aspects need to be improved to help the implementation of a successful approach in future projects. This review provides a tool to facilitate future interdisciplinary research on integration of local and scientific knowledge and will help to devise more successful strategies to tackle the challenges posed by climate change to smallholders in developing countries.

**Keywords:** indicators; local knowledge; soil quality; integration; women; participatory techniques; smallholders

## 1. Introduction

It is becoming increasingly evident nowadays that nearly everywhere on our planet, climate change is happening even faster than most of the scenarios considered in IPCC (Intergovernmental Panel on Climate Change) reports [1,2]. All the rates concerning the impacts of climate change are rising [3], and these impacts and the corresponding risks are related to rising temperatures in a non-linear way [4]. In the years from 2015 to 2019 the average growth rate of the $CO_2$ concentration in the atmosphere was 18% higher than that of the previous 5 years period [3]; temperatures were $1.1 \pm 0.1\,^\circ$C warmer than the pre-industrial ones (1850–1900), and $0.21 \pm 0.08\,^\circ$C warmer than in the previous five years [3]. In all continents this was the warmest post-industrial period registered since systematic scientific climatic observations began [4]. The agreed threshold of $1.5\,^\circ$C has already been

exceeded and experienced by 20–40% of the global population [4], in at least one season. The average rate of sea level rise was 1–2 mm/yr at the beginning of the past century, 3.04 mm/yr one century later, and 4.36 mm/yr in the period 2007–2016 [3]. Glaciers are melting at a rate of −978 mm water equivalent per year: again the most negative rate ever [3]. Extreme events such as intense rains, flash floods and droughts are increasingly frequent, influencing strongly land degradation and soil erosion [3,5]. Coupled with the unfavorable changes in rainfall patterns, enhanced climate variability and pest outbreaks, these effects cause negative impacts on food production, with outcomes depending on land management strategies [3,5]. Moreover, land degradation itself influences climate change, since it causes the release of $CO_2$ in the atmosphere. Soils lose to the atmosphere from 20 to 60% of their organic carbon content when cultivated [5], but other greenhouse gas emissions are related to soil management. These factors account for 21–37% of the global greenhouse gases emitted: among these, 9–14% are due to crop and livestock activities and 5–14% to land use and changes in land use [5]. For all these reasons, soil use is a driver of climate change, but a correct management of soils could offset 5–20% of current global anthropogenic greenhouse gases emissions [5].

Prevention of soil erosion is an important issue, because agricultural productivity is affected by soil erosion in many ways [6]. When soil is washed away by heavy rains or wind, a process exacerbated by many agricultural activities, the physical and nutritional support for plant growth is lost [6]. In fact, the upper layer of the soil, which is eroded first, is also the one that has the largest content of soil organic matter and nutrients; hence its removal causes a loss of fertility [6]. Organic matter has, indeed, several important functions related to soil fertility: It enhances formation of stable soil aggregates, increasing soil porosity, which facilitates the penetration of roots and water drainage [6].

Climate- and soil-related stresses, together with non-climate stressors like population growth, exert an unprecedented impact on food security, playing a key role in recent rises in global hunger [3,5]. Many studies predict that the effects of climate change on crop productivity, crop suitability and grazing systems [7] will be increasingly detrimental, especially at lower latitudes, depending on the management system [5]. More vulnerable to these impacts are those people whose lives depend directly on natural resources, such as smallholder farmers, who already cope with precarious livelihood conditions, particularly those who already depend on degraded lands (1.5 billion worldwide) [5]. Smallholder farmers produce 80% of the food supply in developing countries [8], while family farmers produce more than 80% of the world's food supply [9]. They are more prone to suffer from climatic events, which easily plunge them into poverty, food insecurity, migration, conflicts and loss of cultural heritage [5].

Sustainable soil and environment management can reduce and even reverse land degradation, helping at the same time communities to adapt to and mitigate climate change effects and increase resilience [5]. Agroecology enhances agrobiodiversity (the more diverse, the more resilient), buffers climate extremes, improves ecological processes reducing outbreak of pests and diseases, and delivers ecosystem services [5]. These positive effects can even lead to an increased yield: this is called sustainable crop production intensification [8]. Examples of sustainable management practices are animal integration, SOM (soil organic matter) management, water conservation strategies.

A combined use of all these strategies is crucial to counteract climate change impacts. However, projects aimed at implementing or enforcing sustainable soil and environment management practices in vulnerable communities cannot set aside indigenous local knowledge. Adaptation strategies must be tailor-made and are inseparable from the cultural background, because of the site-specific feature of climate change impacts, environmental characteristics, socio-economic conditions and the linkage between culture, beliefs and food production and consumption [5]. For these reasons, adaptation strategies are successful only when they are identified and implemented through a bottom-up approach, using participatory approaches to involve local stakeholders [5]. Top-down approaches are not effective alone, because they do not gain the support of local people, since they do

not meet their interests; plus they lack the rich local knowledge about complex environmental interactions [10], and the indigenous holistic view of community and environment. Inherited knowledge may be a key factor in facing the new challenges posed by climate change issues because it historically developed specific land-use solutions to site-specific challenges [5], adapting autonomously to climate variability [11].

On the other hand, some cultural beliefs and values may represent barriers to adaptation [5] and sometimes inherited knowledge cannot keep pace with modern sociocultural and economic dynamics, not to mention the challenges posed by threats imposed by climate change. Scientific knowledge can help farmers to take better decisions [10], but these must be integrated into and not cancel out the know-how acquired by generations of farmers.

To facilitate cooperation between researchers, international and local organizations and farmers, therefore, it is essential to build an integrated knowledge base [5]. The first step to allow this integration is the creation of a common language that can be used by all stakeholders to understand each other and reach an efficient cooperation level. This language should be used to discuss climate change impacts on the territory and soil quality and fertility, using quantitative and qualitative indicators that integrate traditional background knowledge with recognized scientific connotations.

In developing countries, women have a fundamental role in agriculture and represent the 43% of labour [12]. Nevertheless, women own only 10–20% of lands [12] and have less access to productive resources (land tenure, inputs, extension services, . . . ), consequently they produce less [12]. It is estimated that if the access to resources was shared equally between men and women, women's productivity potential would be unlocked and the yield in farms would increase by 20–30%, reducing the number of hungry people by 12–17% [12]. Furthermore, women are those who spend more on food, health, clothing and education for their children, so this enhancement would be largely employed for the health, nutrition and education of new generations [12].

For the aforementioned reasons and considering the fact that women are more emotionally and rationally sensitive to issues concerning the future of their children, it is always important to promote gender-sensitive actions, but particularly so in projects aimed to promote climate change actions. Women should be encouraged to participate, not only in discussions, but also in decision-making and planning [12]. Gender issues can be understood better by collecting sex-disaggregated data [12].

In this review, scientific literature has been investigated to gain information about how the problem of the participated building of simple indicators to assess the quality and fertility of soils, has been tackled worldwide in projects devoted to rural communities of developing countries. Our goal was to find out how experience gained from these studies can help researchers to implement projects aimed at counteracting enhanced risks of food insecurity and poverty caused by climatic changes in developing countries.

The aim of this review is to provide answers to the following questions:

1. Which local indicators have so far helped rural communities to acquire awareness of management impacts on the fertility of soils?
2. Which technical indicators have been used by researchers to assess the quality and fertility of cultivated soils?
3. Which approaches have been used for the integration of local and scientific knowledge?
4. Which participatory techniques have been applied in these contexts?
5. Has there been any involvement of women in these studies?
6. What are the reasons for the choices operated by researchers and what are the methodological shortcomings that should be dealt with in future studies?

To this purpose we performed a meta-analysis of papers reported in the scientific literature, examining bibliographic aspects of publications, expertise of involved teams and tools employed (local and technical indicators and participatory techniques) in soil evaluation.

The outcomes of this research can represent a helpful introductory working tool for the definition of soil conservation guidelines that combine inherited knowledge and modern technical expertise.

## 2. Materials and Methods

### 2.1. Literature Search Methodology

Our literature search started with a preliminary investigation on the main keywords chosen by papers published on the subject of participatory sustainable soil management from 1990 to the present day. The search was carried out on the 29 July 2020, using the database Scopus and the keywords selected were: ("soil quality" OR "soil fertility" OR "soil management" OR "agricultural sustainability") AND indicator* AND participatory. A first screening using the keywords "soil quality" OR "soil fertility" OR "soil management" OR "agricultural sustainability" in order to select papers focused on soil management, returned 44,983 papers. Narrowing down the search and selecting only papers containing the term "indicator*", 4791 papers were obtained. Among them, papers that used participatory approaches (contained the keyword "participatory") were only 70. Among these, some papers were further excluded based on the following exclusion criteria: the paper must be available to the general public; the paper must report results of studies carried out in developing countries; the use of participatory approaches used must be at least mentioned.

The papers possessing these characteristics (43) are identifiable in the References by the symbol @. The papers were examined following some guide questions: which participatory approaches have been used? Does the paper talk about local indicators? If so, what are they assessing? Do they address climate change? Or do the authors mention climate change in any way? Which are the local indicators used? Are the authors trying to find an integrated approach for local and technical/scientific knowledge? If so, which is the methodology? Which are the technical indicators used? If no integration is sought, how was the information collected thanks to local knowledge used?

### 2.2. Bibliographic Analysis

The papers selected refer to projects implemented in 23 different countries, distributed among three continents. The geographical distribution is described in Figure 1 and shows that most papers are set in Africa. This could be a random result or could indicate that participatory approaches are more frequently used in African projects.

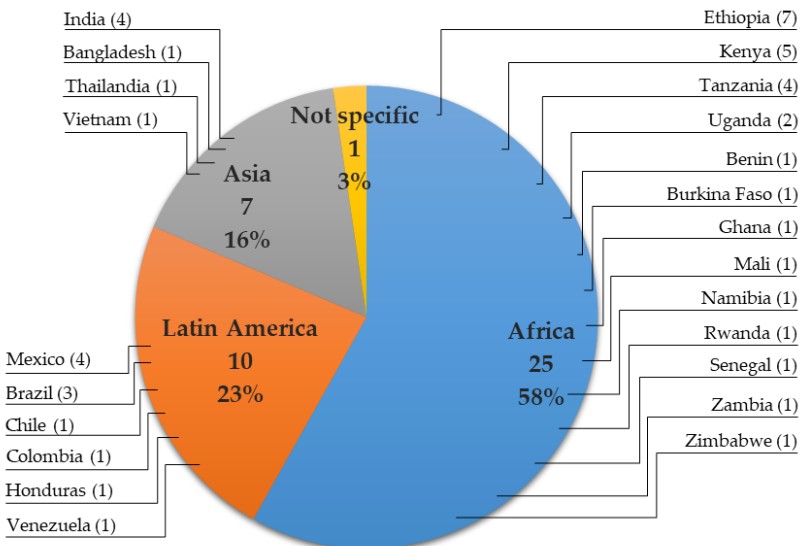

**Figure 1.** Geographical distribution of papers.

The most frequent journals' main topics are (Figure 2b): environment (the majority of them dealing with soil issues), agriculture and environment together or agriculture alone. This derives from the fact that the projects that use participatory approaches in rural places of developing countries are focused on helping farmers to maximize crop production without overexploiting the soil and the environment. It is noteworthy that, despite the fact that participatory approaches are based on social principles, there are no papers on the subject published in journals dealing with social sciences; only two articles have appeared in journals in part related to social sciences, precisely to ethnobiology. This is surprising, considering how important it would be to develop guidelines for a scientifically sound sociological approach to participation.

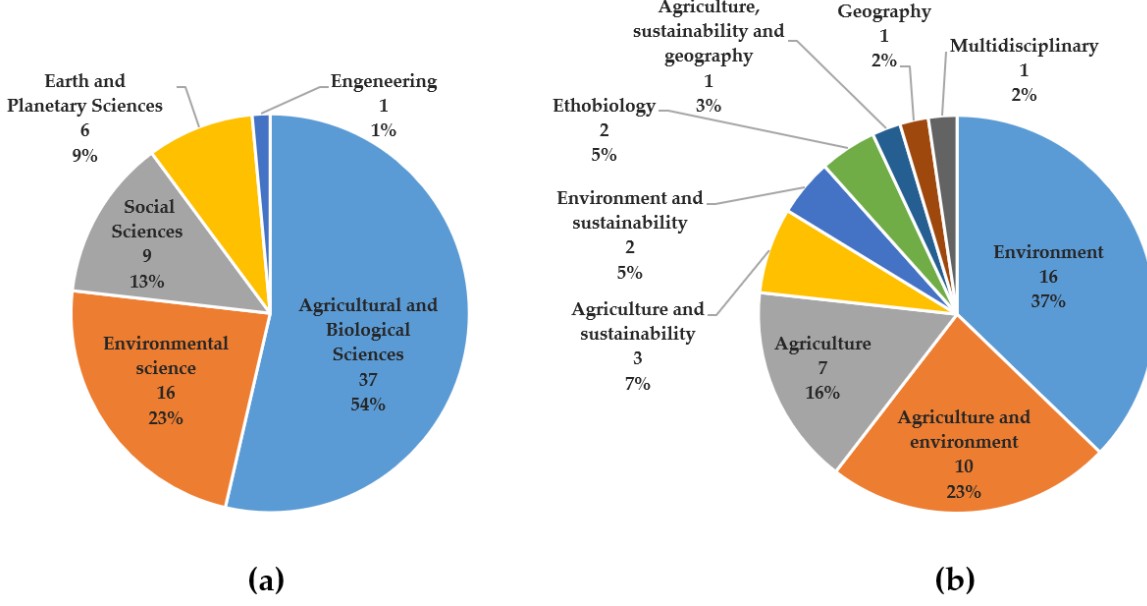

**Figure 2.** (**a**) Author main experience field following Scopus indications. (**b**) Number of papers for every journal topic.

Finally, the composition of the research groups was examined, using the indication given by Scopus about every author's main subject area (Figure 2a). This information can be found opening the author sheet and selecting the voice "show all authors info" under the title "Subject areas". Most papers (19) were written by research groups of joint agricultural and biological sciences experts. They collaborated with environmental experts in 6 papers. Papers written by environmental experts only (5) were published on environment, sustainability and geography journals. Earth and planetary science experts worked together with environmental, agricultural and biological sciences experts and published on environmental journals. The only group featuring an engineer was composed of agricultural and biological sciences experts and published on a multidisciplinary journal [13]. Only 9 papers were written in collaboration with social sciences experts [14–22], who worked in multidisciplinary teams (6), or only with agricultural and biological sciences experts (3) and published in all types of journals.

*2.3. Correlation Analysis*

The correlation among all the characteristics of the paper studied was calculated using the qgraph tool with R software which calculates Pearson correlation coefficients among factors. There are different ways to represent data, and the one chosen shows different levels of a factor represented with circles and grouped in clusters. The circles are connected by green and red lines, which indicate positive and negative correlations, respectively. The width of a line corresponds to the absolute weight and scale relative to the strongest weight in the graph. Lines with an absolute value under the minimum argument (0.5 in this case) were omitted. The values are reported in the Appendix A.

## 3. Discussion

A list of all technical and local indicators mentioned in the papers is reported in Tables 1 and 2. We reported all the indicators in order to give an idea of their number and large variability among local indicators. To allow a comparison, they have been grouped using the same categories: physical properties, chemical properties, organic matter, bioindicators, hydraulic properties, mechanical properties, management, workability, input, indexes.

**Table 1.** List of all technical indicators mentioned in the selected papers. As regards the DPSIR (Drivers-Pressures-State-Impact-Response) classification, all indicators are considered State-Impact indicators, except for the management, workability, input and indexes categories, where the respective DPSIR classification is indicated in brackets (P = Pressure, R = Response).

| Indicator | | Description | Linked Properties | N° Papers | Bibliography |
|---|---|---|---|---|---|
| PHYSICAL PROPERTIES (26) | | | | | |
| | Color | Visual evaluation of diversity | Soil organic matter (SOM), soil temperature | 1 | [23] |
| Texture (17) | Texture | Weight composition of the mineral fraction of the soil with respect to sizes classes of particles such as sand, silt and clay | Soil physical properties, retention of nutrients and availability of water | 17 | [13–15,17,23–35] |
| | Water dispersible clay | Amount of clay dispersed by shaking soil with water | Resistance to erosion, aggregate stability, formation of crusts | 1 | [30] |
| | Type of clay | Type of clay minerals | Soil physical properties, retention of nutrients | 1 | [23] |
| | Stoniness | Quantity, size, surface exposure of stones | Aeration, permeability, workability, erosion | 1 | [23] |
| Structure (10) | Bulk density | Weight of soil in a given volume | Availability of water, root growth, aeration, porosity, compaction | 7 | [10,13,25,27,32,36,37] |
| | Structure (aggregation) | Presence of aggregates formed by mineral particles and the further association of those aggregates into larger units | Water infiltration, drainage and gas exchange, penetrability of roots, porosity | 3 | [10,13,23] |
| | Total porosity | Volume occupied by voids | Water availability, aeration | 2 | [25,36] |
| | Dispersion of aggregates | Dispersion of soil aggregates in water | Drainage, air exchange, susceptibility to erosion | 1 | [38] |
| | Stability of aggregates | Ability of soil aggregates to resist dispersion | Resistance to erosion | 1 | [38] |
| | Compaction | Lack of porosity | Porosity, water infiltration, drainage and gas exchange | 1 | [39] |
| Depth (3) | Soil depth | Depth to unaltered parent material; the layer that stores water and nutrients for plants | Amount of water, nutrients and space available for root growth. | 1 | [23] |
| | Effective soil depth | Depth to a hard barrier that cannot be penetrated by roots | Amount of water, nutrients and effective space that can be explored by roots, plough depth | 1 | [10] |
| | Horizon thickness | Thickness of a homogeneous soil layer | Amount of water, nutrients, plough depth | 1 | [16] |
| Temperature | | Solar irradiation, heat capacity, orientation | Seedling emergence in spring, evaporation of moisture | 1 | [10] |

**Table 1.** *Cont.*

| Indicator | Description | Linked Properties | N° Papers | Bibliography |
|---|---|---|---|---|
| Slope | Direction and steepness of the field | Infiltration, erosion | 3 | [23,35,40] |
| Erosion | Soil loss | Loss of mass and functionality | 4 | [23,29,39,41] |
| Electrical conductivity (EC) | Amount of salts (indicator of) | Nutrient availability and loss, soil texture, available water capacity | 9 | [13,15,20,23, 24,28,30,32, 37] |
| CHEMICAL PROPERTIES (21) | | | | |
| pH | Measure of soil acidity or alkalinity | Depression of crop yields, availability of nutrients | 19 | [10,13–17,24– 29,31– 34,36,37,40] |
| Available phosphorus (available P) | Fraction of total P in soil that is readily available for absorption by plant roots | Availability of this element often limits crop yields and is not related to total content in soil | 16 | [13–17,24– 29,32– 34,36,40] |
| Total nitrogen (TN) | Macro-nutrient | Crop yield limiting factor | 10 | [13,17,24– 29,33,36] |
| Mineral N | Mineral fraction of total nitrogen | N form ready for crop uptake | 2 | [27,33] |
| Organic N | Organic fraction of nitrogen | Soil N reserves and crop production | 1 | [14] |
| Cation exchange capacity (CEC) | Total capacity of a soil to hold cations in an exchangeable form, measured in an alkaline or neutral buffer solution | Potential availability of nutrients, maximum capacity to retain ammonium, potassium and other micronutrients | 8 | [13–15,17,24, 25,32,40] |
| Effective cation exchange capacity (ECEC) | Total amount of exchangeable cations measured at the soil pH | Availability of nutrients, capacity to retain fertilizers | 1 | [27] |
| Specific exchangeable cations | Amount of a given cation held in exchangeable form ($Ca_2^+$, $Mg_2^+$, $Na^+$, $K^+$) | Crop nutrition status, need of fertilizers | 8 | [13,14,24,25, 27,28,32,33] |
| Potassium (K) | Concentration of K in the soil | Macronutrient for plant growth | 8 | [14,15,17,24, 26,29,32,40] |
| Available K | Fraction of total K in soil that is readily available for absorption by plant root | Nutrient availability | 3 | [13,32,36] |
| Calcium (Ca) | Concentration of Ca in the soil | Enhances clay flocculation, therefore soil aeration and drainage | 7 | [15–17,24,29, 32,40] |
| Base saturation or Exchangeable bases | Percentage of CEC occupied by bases (all cations except $Al^{3+}$ and $H^+$) | Availability of nutrients, capacity to retain fertilizers, correction of soil pH | 6 | [15,17,25,30, 32,36] |
| Magnesium (Mg) | Content of Mg in the soil | Essential element for plant growth | 6 | [15– 17,29,32,40] |
| Total carbon (total C) | Sum of both organic and inorganic C | C sequestered in the soil, soil organic matter | 4 | [10,13,17,27] |
| Micronutrients | Content of Zn, Mn, Cu, Fe, B in the soil | Micronutrients for plant biochemical processes | 4 | [13,24,32,34] |

**Table 1.** *Cont.*

| Indicator | Description | Linked Properties | N° Papers | Bibliography |
|---|---|---|---|---|
| Exchangeable acidity (exchangeable Al or Al and H) | The amount of acid cations, Al and H which occupy the CEC | Correction of soil pH | 3 | [24,27,33] |
| Na | Content of Na in the soil | Soil aggregation and aeration | 3 | [15,29,32] |
| Available N | Mineralizable N. | Nutrient availability | 2 | [13,32] |
| C/N ratio | Ratio of the mass of carbon to the mass of nitrogen | Degree of organic matter transformation | 2 | [14,33] |
| Inorganic C | C in carbonates | Availability of nutrients, correction of soil pH | 1 | [30] |
| Fe and Al (hydr-) oxides | Concentration of Fe and Al oxides and hyroxides | Organic carbon (OC) protection, nutrients availability | 1 | [17] |
| ORGANIC MATTER (22) | | | | |
| Organic carbon (OC)/organic matter (OM) | Fraction of organic C in the soil, contained in biota and biotic material | Soil structure, aggregation, water retention, biodiversity, pollutants absorption and retention, buffering capacity, nutrients, fertility, cation exchange capacity | 22 | [13–17,20,23–37,40] |
| Active Carbon (AC) | Energy source for soil microorganisms | Biological activity | 1 | [28] |
| BIOINDICATORS (7) | | | | |
| Soil fauna | Soil macro- and micro fauna | Biological soil health | 4 | [10,23,32,37] |
| Microbial biomass C | C present in soil as microbial biomass | SOM quality and dynamics | 3 | [13,37,41] |
| Dehydrogenase activity | Enzyme for biological oxidation of SOM | Overall soil microbial activity | 3 | [13,32,37] |
| Vegetation | Local plant species | Biological soil health | 2 | [10,23] |
| Alkaline Phosphatase activity | Enzyme that releases phosphate | Levels of microbial activity | 2 | [32,37] |
| Total bacteria | Bacterial biomass in soil | Biological soil health | 1 | [37] |
| Microbial biomass | Total amount of microrganisms in soil | Biological soil health | 1 | [10] |
| Microbial biomass N | N concentration in microbial biomass | SOM quality and dynamics | 1 | [37] |
| Soil respiration | Production of $CO_2$ by soil organisms (roots, microbes, fauna) | Nutrients cycling, microbial activity, SOM content and decomposition | 1 | [10] |
| Rates of litter decomposition | Rate of organic material decomposition into prime constituents | Nutrients cycling | 1 | [10] |
| Fluorescein di-acetate | Measure of global hydrolysis capacity of soil | Soil biological capacity | 1 | [32] |
| Plant uptake ratio | Ratio of N and K uptake respect to P uptake | Plant-soil interactions | 1 | [14] |

**Table 1.** *Cont.*

| Indicator | Description | Linked Properties | N° Papers | Bibliography |
|---|---|---|---|---|
| HYDRAULIC PROPERTIES (7) | | | | |
| Water content | Soil moisture at sampling | Water availability for soil organisms and plants | 2 | [10,30] |
| Available water content/capacity (AWC) | Amount of water that a soil can store that is available for use by plants | Water availability for plants | 2 | [13,25] |
| Water holding capacity | Ability of a certain soil to hold water against the force of gravity | Water availability for plants | 2 | [23,32] |
| Soil moisture retention | Water retained by the soil | Water availability for plants | 2 | [13,29] |
| Field capacity (FC) | Soil water content after excess water has drained away | Water availability for plants | 1 | [25] |
| Permanent wilting point (PWP) | Soil moisture at which there is no available water for plants | Water availability for plants | 1 | [25] |
| Infiltration | Soil's ability to allow water movement into and through the soil profile | Water availability for plants, erosion | 1 | [23] |
| Drainage | How rapidly excess water leaves the soil by runoff or internal drainage | Water availability for plants, erosion | 1 | [23] |
| MECHANICAL PROPERTIES (3) | | | | |
| Penetration resistance (PR) | Difficulty to penetrate soil with penetrometer | Plough difficulty, roots penetration capacity | 2 | [30,38] |
| Effective rooting depth | The soil depth from which a fully grown plant can extract water and nutrients | Ecosystem resilience to drought | 1 | [10] |
| MANAGEMENT (8) | | | | |
| Plant density (R) | Number of plants per square meter | | 2 | [35,40] |
| Crop performance (R) | General evaluation that can include yield, growth rate, crop status | | 1 | [29] |
| Field type (P) | Classification based on production activities, resource allocation and management practices | | 1 | [33] |
| Cropping history (P) | Record of the crops cultivated on the land in the previous years | | 1 | [10] |
| Intensity of crop rotation (P) | Years before a crop is planted again in the same field | | 1 | [39] |
| Diversity of crop rotation (P) | Number of different crops in a rotation | | 1 | [39] |
| Sowing date (P) | The day of sowing | | 1 | [35] |
| Agro-biodiversity (P) | Species of crops, domesticated animals and multipurpose trees per acre | | 1 | [20] |
| Intercropping practices (P) | Multiple cropping practice that involves growing two or more crops in proximity | | 1 | [29] |
| Improved fallow (P) | Land resting from cultivation with planted species of leguminous trees, shrubs and herbaceous cover crops | | 1 | [29] |
| Continuous cropping (P) | Cultivation of the same type of crops on the same piece of land every year with the absence of protracted fallow periods | | 1 | [29] |
| Contour farming operations (P) | Seeding and spraying on crops planted across or perpendicular to slopes to follow the contours of a slope of a field | | 1 | [39] |

**Table 1.** *Cont.*

| Indicator | Description | Linked Properties | N° Papers | Bibliography |
|---|---|---|---|---|
| Agroforestry (P) | Land use management system in which trees or shrubs are grown around or among crops or pastureland | | 1 | [29] |
| Time of harvesting (P) | Harvesting date | | 1 | [29] |
| Weed management (P) | Weed control operations | | 1 | [29] |
| Tillage depth (P) | Centimeters of soil interested by tillage | | 1 | [29] |
| Frequency of soil tillage (P) | Years of interval between tillage operations | | 1 | [39] |
| Time of adoption of the no-tillage system (P) | Number of years after application of a no-tillage system | | 1 | [39] |
| Water management (P) | How irrigation is performed | | 1 | [37] |
| Trenches (P) | Soil digs to enhance water infiltration and prevent soil erosion | | 1 | [29] |
| Agricultural terraces(P) | Soil management technique to reduce soil erosion and make cultivation easier | | 1 | [39] |
| Burning of bushes (P) | Traditional practice of burning bushes | | 1 | [29] |
| WORKABILITY (6) | | | | |
| Yield (R) | Amount of an agricultural product harvested per unit of land area | | 6 | [10,14,23,29, 35,40] |
| Fertility (P) | The ability of soil to sustain plant growth and optimize crop yield | | 1 | [23] |
| Primary productivity (R) | Rate at which plants and other photosynthetic organisms produce organic compounds in an ecosystem | | 1 | [40] |
| Presence of physical barriers (P) | Presence of rocks or other impediments | | 1 | [23] |
| Lack of knowledge and skills on soil fertility management (P) | Evaluation of the gaps in knowledge and skills concerning soil fertility management | | 1 | [29] |
| Pests and disease incidence (P) | Type and frequency of pest and disease incidence | | 1 | [29] |
| Drought (P) | Incidence of drought events | | 1 | [29] |
| INPUTS (8) | | | | |
| Resource flows (R) | Input, output and internal flows of nutrients at farm level | | 3 | [14,18,42] |
| Fertilizer management (P) | Quantity and type of fertilizer used | | 1 | [37] |
| Fertilizers cost (P) | Cost for the purchase of inorganic fertilizers | | 1 | [29] |
| Farmyard manure (P) | Stabilized organic fertilizer from animal feces and straw | | 1 | [29] |
| Green manure (P) | Crops grown for the express purpose of ploughing them in | | 1 | [29] |
| Organic fertilizer (P) | Fertilizer created from animal or vegetable matter, animal or human excreta | | 1 | [35] |
| Compost (P) | Decomposed organic matter | | 1 | [29] |
| Mulching (P) | Layer of material applied to the surface of soil | | 1 | [29] |
| Phosphorous fertilizer (P) | Different types and quantities of P fertilizers | | 1 | [29] |
| Residue load (P) | Quantity of residues | | 1 | [37] |
| Residue management (P) | Percent of residue incorporated in soil | | 1 | [37] |
| Incorporation of organic residue (P) | Incorporation of organic residues in the soil | | 1 | [23] |
| Persistence of crop residues on the soil surface (P) | Amount of undecomposed plant remains | | 1 | [39] |
| Management of available organic materials (P) | How organic manures are used (if discarded or incorporated) | | 1 | [29] |

**Table 1.** *Cont.*

| Indicator | Description | Linked Properties | N° Papers | Bibliography |
|---|---|---|---|---|
| | | INDEXES (8) | | |
| Nutrients flows and balances (R) | Calculated flows and balances of nutrients in the soil or agricultural system | | 8 | [10,13,14,18, 35,39,41,42] |
| N use efficiency (R) | Relationships between the total N inputs and outputs | | 2 | [35,41] |
| Incremental response to applied N (R) | Yield increase with increasing fertilizer input | | 1 | [35] |

**Table 2.** Local indicators used in the selected papers. The numbers are referred to the total number of papers which reported the use of that specific local indicator. For every type of indicator, all different ways of identifications are reported and, where specified, the assessment method. When there is no specification, it means that the general definition has been used, without further explanations. The letters P and R in brackets represent Pressure and Response indicators for DPSIR classification (further explained).

| Category | N° | Local Indicator Class | Denomination | Bibliography |
|---|---|---|---|---|
| PHYSICAL PROPERTIES (21) | 17 | Color | fertility, compaction | [10,14–16,23–28,31,36,43–47] |
| | 14 | Texture | Texture | [10,15,26,27,33] |
| | | | Presence of clay, silt, sand, gravel | [15,21,23,27,28,47] |
| | | | Presence of clay, silt, sand, gravel by touch | [43] |
| | | | Presence of cracks (clay type) | [14,23,45] |
| | | | Dustiness | [10,43] |
| | | | Stickiness | [21,23,43] |
| | | | Sponginess | [23] |
| | | | Heaviness/lightness | [45,47] |
| | | | Stoniness | [14,15,21,23,28,36,47,48] |
| | 14 | Field location | Field location | [15,28] |
| | | | Slope | [10,16,21,23,24,33,36,43–45,47,48] |
| | | | Altitude (climate) | [15] |
| | | | Respect to spontaneous vegetation | [21,47] |
| | | | Respect to grazing lands | [47] |
| | | | Distance from homestead | [33] |
| | 13 | Structure | Structure | [10] |
| | | | Compaction | [10,14,45,47] |
| | | | Hard pan on soil surface | [27,28,47] |
| | | | Aggregation | [27,28] |
| | | | How soil pieces break in hands (aggregation) | [28] |
| | | | Dimensions of pieces after shovel | [28] |
| | | | Dispersion of aggregates | [38] |

**Table 2.** *Cont.*

| Category | N° | Local Indicator Class | Denomination | Bibliography |
|---|---|---|---|---|
| | | | Clumping | [10,28] |
| | | | Consistence (Looseness/softness/hardness) | [10,15,23,27,31,43,47] |
| | | | Porousness | [10,21] |
| | | | Plasticity | [15] |
| | | | Friability | [24] |
| | | | Powdery | [28] |
| | | | Erosion | [45] |
| | | | Washed soil | [23,28] |
| | | | Laminar erosion | [26,47,48] |
| | | | Water ways (Gullies and rills) | [21,26,44,46–48] |
| | | | Eroding clods | [48] |
| | 10 | Erosion | Splash pedestals | [47,48] |
| | | | Presence of cracks | [14,23,45] |
| | | | Sediment deposits | [47,48] |
| | | | Build-up of soil against barriers | [47] |
| | | | Soil blown away by wind | [28,47] |
| | | | Soil sloping | [28] |
| | | | Rocks exposure | [44,47,48] |
| | | | Root exposure | [44,47,48] |
| | | | Subsoil exposure | [47] |
| | | | Crop seedling removal by water | [44] |
| | 7 | | Depth/thickness | [10,15,21,23,26,45,47] |
| | 1 | | Smell (decaying/sour/fresh/earthy/absent) | [28] |
| | 1 | | Soil temperature | [28] |
| ORGANIC MATTER (8) | 5 | | Fertility | [23,33,43,45,48] |
| | 2 | | Soil Organic Matter | [15,21] |
| | 1 | | Soil Organic Matter by touch | [43] |
| | 1 | | Black layer (Soil Organic Matter) | [23] |
| | 1 | | Humus layer | [16] |
| HYDRAULIC PROPERTIES (17) | 7 | | Water logging/Ponding flooding | [16,21,28,31,43,45,47] |
| | 6 | | Moisture holding capacity/content | [15,16,21,27,36,47] |
| | 5 | | Water Holding Capacity | [10,24,26,33,36] |
| | 4 | | Water retention | [23,31,36,43] |
| | 3 | | Drainage rate (visual) | [23,43,47] |
| | 2 | | Runoff | [16,48] |
| | 1 | | Water availability | [46] |
| | 1 | | Water content | [10] |
| | 1 | | Moisture (by touch) | [43] |
| BIOINDICATORS (17) | 10 | | Local plants (presence and status) | [10,14,15,21,23,24,26,28,43,46] |

Table 2. *Cont.*

| Category | N° | Local Indicator Class | Denomination | Bibliography |
|---|---|---|---|---|
| | 10 | | Weeds | [10,24–27,33,36,45–47] |
| | 10 | | Soil fauna | [10,14,16,23,26–28,36,43,47] |
| MECHANICAL PROPERTIES (6) | 4 | Salinity | Salinity | [15] |
| | | | Presence of salts | [14] |
| | | | White crust | [28] |
| | | | Water becoming salty | [46] |
| | 1 | | Penetration resistance (compaction) | [38] |
| | 1 | | Roots penetration (compaction) | [28] |
| | 1 | | Shovel penetration resistance (hardness, compaction) | [28] |
| | 1 | | Carbonate concretions | [45] |
| MANAGEMENT (10) | 7 | Management techniques (P) | Plow depth | [21,25,27,47] |
| | | | Machines used | [21] |
| | | | Soil burning | [23] |
| | | | Frequency of watering (Water Holding Capacity) | [14,28] |
| | 3 | Cropping (P) | Cropping history | [10] |
| | | | Fallowing history | [10] |
| | | | Crop varieties (qualitative) | [21] |
| | | | Crop rotation | [21] |
| | | | Territorial composition | [21] |
| | | | Land use/coverage | [15] |
| | 3 | Protection techniques (P) | Windbreaks | [21] |
| | | | Terraces | [44,47] |
| | | | Edge/contour/bunds | [21,44] |
| | | | Grass strips (for stabilization) | [44] |
| | | | Ditches/water ways/drains/check dams | [21,44] |
| WORKABILITY (18) | 15 | | Yield/productivity (R) | [10,16,21,23–28,31,36,44,46–48] |
| | 11 | Soil workability | Workability/Ease of ploughing (P) | [10,21,23–25,28,36,43,45,47] |
| | | | Need for hoeing (R) | [26] |
| | | | Need for crop rotation (R) | [26] |
| | | | Weed labour (P) | [47] |
| | 11 | Crop vigour (R) | Crop vigour | [14,26,36] |
| | | | Growth rate | [26,36,43] |
| | | | Even/uneven growth | [47] |
| | | | Maturity time | [47] |
| | | | Time to flowering | [10] |
| | | | Crop health | [31] |

**Table 2.** *Cont.*

| Category | N° | Local Indicator Class | Denomination | Bibliography |
|---|---|---|---|---|
| | | | Plant strength | [43] |
| | | | Size | [28,43,47] |
| | | | Establishment | [24,27,28] |
| | | | Coloration | [24,27,28,43,47] |
| | | | Leaf edges "burnt" (salts) | [28] |
| | | | Plants stunted (vigour, salts) | [24,27,28] |
| | | | Misshapen plants | [28] |
| | | | Crop density | [47] |
| | | | Soil-borne diseases (P) | [28] |
| | | | Tree die-off (R) | [46] |
| | 2 | Crop performance (R) | | [14,45] |
| | 1 | Roots status (R) | Roots health | [28] |
| | | | Roots dimension | [28] |
| | | | Roots colour | [28] |
| | | | Galls on roots | [28] |
| | 1 | Versatility (soil holds multiple functions to farmers) (P) | | [28] |
| | 5 | Fertilizer need (P) | | [23,25,28,45,47] |
| | 3 | Fertilizer response (R) | | [27,28,47] |
| | 2 | Inputs need (P) | | [21,47] |
| | 1 | Mineral fertilizers incorporation (P) | | [18] |
| | 1 | Organic manure need (P) | | [26] |
| INPUTS (10) | 1 | Organic residue incorporation (P) | | [23] |
| | 1 | Production and use of organic fertilizers (P) | | [18] |
| | 1 | Livestock manure incorporation (P) | | [43] |
| | 1 | Tree biomass incorporation (P) | | [43] |
| | 1 | Amount of crop residue (P) | | [43] |
| | 1 | Crop residue recycling (P) | | [18] |

### 3.1. Technical Indicators

The majority of studies (65%) made use of technical indicators.

### 3.2. Local Indicators

Local indicators are those used by farmers in their daily life to take decisions about agricultural and management issues. They describe characteristics that can be observed directly, mostly with sight and touch, but sometimes with smell, and by the constant and careful observation of environmental phenomena. They have been studied by researchers in many ways and using different approaches, but always with the collaboration of local populations. The peculiarity of local indicators is that many of them summarize various aspects of soil quality [10], but may sometimes be redundant [26].

In the selection, 22 out of 43 (51%) papers made use of local indicators.

### 3.3. Comparison between Technical and Local Indicators

It is possible to observe (Figure 3a) that indicators related to physical properties are the most widely used in both local and technical knowledge.

Texture and structure are used in both local and technical knowledge, while color is typically used by local knowledge. It is an indicator that depends on many factors, so that allows us to undertake an evaluation of the general quality and fertility of soils, without the distinction between single factors; often local indicators have this multiple function [10,26]. As regards field location, technical indicators refer to slope, while local ones refer to other factors, besides slope, such as the features of the surroundings. Erosion signs are recognized by both scientists and farmers, but are not specified in the papers, while multiple local erosion indicators are reported. Even soil depth is more widely used in local knowledge.

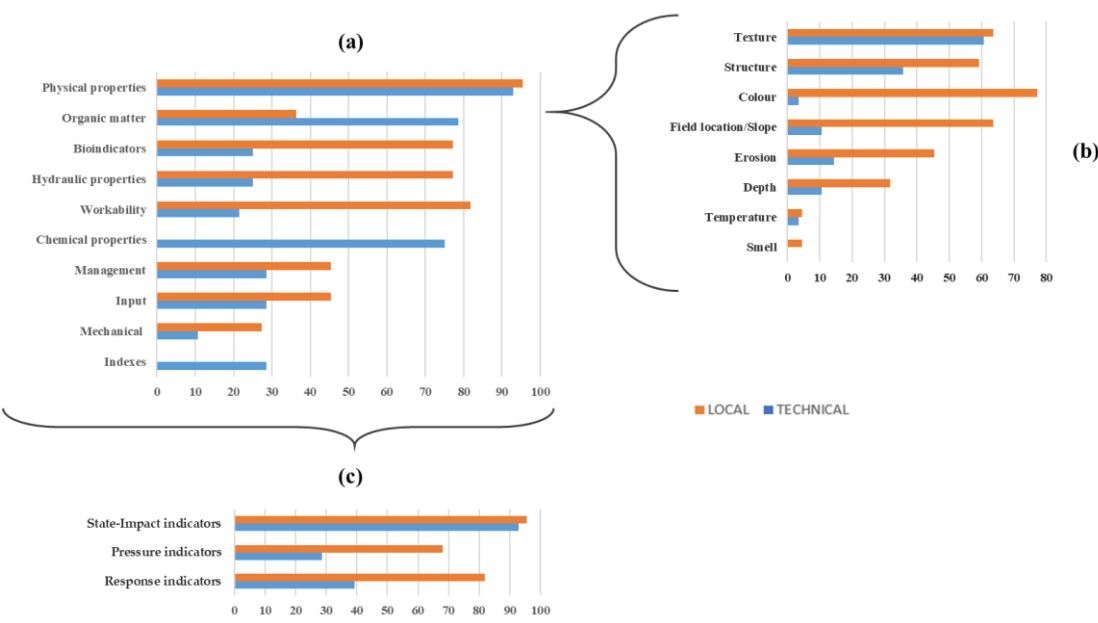

**Figure 3.** (**a**) Percent of papers using certain technical and local indicators, with respect to the total number of papers using technical and local indicators respectively. (**b**) Focus on physical properties. (**c**) DPSIR classification. Percent of papers using State-Impact, Pressure and Response technical and local indicators, with respect to the total number of papers using technical and local indicators respectively.

In local classifications, usually there is no distinction between soil type and level of degradation, but the erosion signs are characteristics used to identify a soil class [49]. Even land suitability enters this classification method [49].

Organic matter is assessed more often by scientists, because even if farmers understand the concept of soil fertility and can evaluate it as a whole, they cannot always understand the importance of soil organic matter alone. Bioindicators are widely used by farmers: they observe the presence of plants and animals in the crops and in the surroundings. Scientists use mainly factors measured through laboratory methods, assessing microbial biomass or presence of organic compounds which indicate biological activity. Hydraulic properties too are more important for farmers. Again, only through sight or touch, they observe how water behave, how rapidly it is absorbed by the soil, if ponds are formed and if superficial flooding happens. Chemical properties obviously belong only to scientific indicators. Among local indicators, the observation of carbonate concretions has been counted as a chemical indicator.

Salinity problems (Figure 3b) are assessed by electrical conductivity (scientific method) or by observing the presence of salts (farmers' method), the latter evidently a less efficient way as it does not allow salinity problems to be detected unless there is a visible efflorescence at the soil surface. Mechanical characteristics are assessed using soil penetration resistance, evaluated by farmers using shovels. Really important indicators for farmers

(less used among scientists) are those directly related to their daily work: workability, management, input. Finally, indexes are only part of scientific assessment methods.

An analysis of the indicators in terms of the DPSIR classification (Drivers-Pressures-State-Impact-Response) (Figure 3c reveals that the ones that describe soil quality are State and Impact indicators, since they describe the state of a soil at a given moment, but they can also be used to highlight any change in soil quality. Management, workability and input indicators are Pressure and Response drivers, as they describe actions that influence the State, or the Response of the system to the Pressures. It is possible to observe that local knowledge uses more frequently Pressure and Response indicators, as they are more useful for agricultural purposes, and because they can be observed and understood without the necessity to understand how the environmental system works from a biological or chemical point of view.

In Tables 1 and 2 all indicators are reported, for the sake of completeness, but many of them are used only in a few papers. The most used technical indicators are: texture for what regards physical properties, pH and available P as chemical properties, organic carbon for organic matter. Among local indicators there is a greater apparent dispersion because there is no general protocol, every farmer acts according to his/her experience. Anyway, the most widely used indicators are color, texture, field location, structure for physical properties, yield and productivity in "workability" category.

Unfortunately, the papers, often do not explain how local indicators were assessed. This not only does not allow access to useful information that could be valuable to other researchers, but impedes progress in the methodology.

*3.4. Integration between Technical and Local Knowledge*

As already mentioned, the integration of local and technical knowledge is essential in sustainable development projects, especially when their aim is to empower communities, instead of delivering finished tools that might not be adopted in practice or abandoned in the long term [10,50]. As reported by Cook et al. [51], farmers prefer to take part to empirical approaches in which their traditional knowledge is central and which involve on-farm experiments, rather than to be given ready-made recipes for soil management and have to deal with scientific knowledge. The attention to local indicators and inherited knowledge allows the real needs and desires of farmers to be targeted, adhering to the farmer's own concept of development, while providing essential information to scientists to understand the root of problems. At the same time, integration is important to make the outcomes of scientific knowledge serviceable to farmers.

As regards the articles reviewed, the general strategy employed was: to acquire local information through participatory approaches, and then try to find a correspondence with scientific notions. Deugd et al. [42], in their theoretical analysis about integrated nutrient management methods, recommend a collaboration in which both researchers and farmers express their perceptions, learn from one another and find an integration between their visions.

Some studies used an ethno-pedological approach: researchers studied the local system of soil classification, including its local nomenclature [16,23,36,43], and eventually compared it to the international classification [15,31,40], from which they were able to derive further information. Conversely, researchers may identify a suitable set of local indicators with which classify soils; they then put to scrutiny the classification thus obtained with the aid of technical indicators [16,26,27,40] or compare it to an alternative classification built with technical parameters obtained through soil analysis [10,14,25]. Another approach starts by studying local perceptions [17,24,25] or local indicators for a given issue [26,36] and puts them directly to scrutiny using scientific concepts. A correspondence between local and technical indicators can be found qualitatively [10,14,23,26,28,43], or quantitatively studying relationships among local and technical indicators and the correlation among them [17,26]. Prudat et al. [31], instead of keeping local and technical indicators separated, used local knowledge and soil analysis results to elaborate a new set of integrated indicators. Defoer et al. and Mowo et al. [14,18] used data collected with participatory

approaches to build models. A different approach was used by De Auraújo [38]: simple experiments were performed on the field by farmers with simple instruments, while soil samples were analyzed in the laboratory, to assess the accuracy of the field method.

Where soil sampling and chemical analyses were carried out, there were different ways to choose the sampling design: in some cases farmers were asked to point out their best and poorest soils [17,25,26,28,36], so sampling decisions were totally up to them. Another approach was to put together information gathered previously by researchers with the outcomes of the participatory activities [15,16,24,27,40]. Finally, in some cases the sampling was, however, entirely designed by researchers following technical notions [14,31,38].

In the cases where quantitative data were present, the comparison between local and technical knowledge was either only qualitative [15,24,28,36], quantitative but using simple instruments [14], or was subjected to a statistical analysis of data [16,17,25–27,31,38,40,43].

### 3.5. Participatory Techniques

Participatory methods are used by researchers, community members and activists to allow people to participate and be influential on the decisions taken in their own communities [52]. They have been used in many different countries and places [52]. They are, indeed, the combined product of many interactions occurred in the past, beginning in the late Seventies, in the field of development cooperation, to collect data from local people and to respond to local necessities [53]. Historically, the first methodology used was the so called Rapid Rural Appraisal (RAR), that considered local people only as a source of information, but did not give them any decision power [53]. Afterwards the Participatory Rural Appraisal (PRA) was created [54]. It gave more focus to the active role played by people and to the reversal of learning, from people to researchers, to the importance to hear every voice, even the outlying ones, seeking diversity. The increasing acceptance from facilitators of the mutual learning concepts led to the introduction of other terminology in the early 1990s: the Participatory Learning and Action (PLA) [53].

A widely used term for this type of implementation strategy is participatory action research (PAR), which defines a general approach that makes researchers and participants work together to understand and solve problems in a context-specific way, to promote social change and democracy. It is characterized by an iterative cycle of research, action and reflection on results which makes use of the array of quantitative and qualitative methods [52].

The participatory methods reviewed are multiple and have been grouped in 6 types, summarized in Table 3. In the same way we have provided all the indicators mentioned, all the terms concerning participation have been reported in Table 3. The nomenclature is vast and not standardized, there are many synonyms or very similar instruments, which have been reported in brackets. The numbers in brackets represent the number of papers in which those types of approach were used.

These techniques were formed in the field of applied anthropology as innovative practices, with the aim to meet the agencies' necessities for the realization of projects: differently from the traditional field, anthropologists had to perform their work in a limited period of time, therefore the necessity of rapid techniques, as focus groups, semi-structured interviews, sorting and ranking, participatory action research techniques [55]. An output of these PRA exercises are the future actions plans, like the Community Action Plan, which consists of the identification of goals, and actions to reach them [52]. Participant observation is a socio-ethnological tool that has been used for long time and consists in the observation and collection of information by researchers that participate in local social life [52].

All the participatory techniques reported in Table 3 can be implemented either allowing the participation of anyone willing to participate or by selecting appropriate groups of "key informants", chosen by farmers, researchers or institutions, based on criteria such as importance in the community, wealth, soil and environmental knowledge, ability to manage their fields.

**Table 3.** Complete list of participatory techniques used in the selected papers.

| Category | Instruments/Synonyms |
|---|---|
| 1—Group activities (38) | Meetings to disseminate and collect information (informal m., initial m., community m.), group discussions (plenary session, formal and informal d., semi-structured d. with check-list, brainstorming sessions, focus groups, conversations between researchers and participating farmers, consultations with stakeholders)<br>Workshops<br>Assessment/diagnosis meetings (soil quality status a., soil quality a./d., soil health a., impact a., rural a.), selection of indicators<br>Ranking sessions (listing and sorting, wealth r., resource r., categorization, prioritization, matrix, pair-wise r., classification of land quality), analysis (data a., cause-effect a., trend a.), scoring. Consensus building, community planning (community action p., annual p. meetings), Dissemination meetings (distribution of reading materials, leaflets, pamphlets written in the common language), quality control, monitoring and evaluation. |
| 2—Field activities (17) | Transect walks (t.w. with check-list, farm transects), farm walks (field observation, field trips, field-days), farm visits (farmer exchange v., study tours, cross-site v.), field monitoring, field activities<br>Mapping (farm m., soil type m., soil fertility m., resource flow m.)<br>Photo observation (p. elicitation) |
| 3—Interviews (27) | Surveys (household., diagnostic., reconnaissance., field., follow-up.), Interviews (key informant i., in-depth i., group i., focus group i., individual i.),<br>Questionnaires (semi-structured q., structured q., comprehensive q.) |
| 4—Training sessions (5) | Trainings, workshops |
| 5—Fairs (2) | Soil fairs |
| 6—Experiments (9) | On-farm testing, farmer′s research trial/experiment (on-farm t., fertilization e., Mother-baby t.), participatory testing, technology development |

Group activities can be used at all stages of the project. The main instrument is group discussion, by which people are encouraged to talk to each other, with the help of two mediators: a facilitator and a note-taker [52]. The objectives are multiple: collect information, select key informants, involve stakeholders, build consensus, identify indicators and objectives, rank them, take decisions with stakeholders, draw maps, evaluate the results of the project. The peculiar feature of this category is that people are made to work together, and everyone is supposed to be an active subject in the process, but the work is not carried out in the field. Meetings are used to communicate with everyone. They can be specifically organized for the purpose, or be part of normal community meetings and used to communicate with all community members.

Workshops are, on the contrary, group activities in which a small group, helped by a facilitator, explores defined issues, develops ideas and takes decisions [56].

In meetings devoted to ranking, a wide variety of ranking and scoring exercises are used to weight different options against different criteria [57] and help farmers to identify indicators and to create a classification based on them. A ranking session can begin with a discussion, in which participants prepare a list of all available options. Afterwards, participants must select the preferred option, which is ranked one, and so on. The facilitator should help to start the discussion and explain the method, but he/she should remain only an observer during the analysis. In the end, participants must also explain the reasons for their choice. The act of scoring is a bit different, because options are given a score and not only a rank, so that it is possible to weight differences among the different options [57].

In the examined papers different exercises of mapping and modelling were also adopted. They were carried out with GIS (Geographic Information System) instruments and verified on the spot with the help of farmers: the important thing is to let even non-literate people express their knowledge about the place where they live [52]. Instruments, including simple ones like sticks, are usually used as a starting point, since they help everyone to understand [54] and are useful as icebreakers [57]. The elements mapped are

the farms [23], the resource flow [14,18,33,35], the land use [16,17], the soil types [17,31,43], the soil fertility [14] and the social conditions [42].

One paper only [46] used photo elicitation: pastoralists were given cameras to take photos, which were afterwards discussed during focus groups, leading to the identification of local indicators.

The terms assessment and diagnosis are used in a broad sense, since they can represent the final aim of the other techniques described above, or they can be used where the method is not described.

Field activities are a direct way to collect and convey information and mostly consist in field visits or transect walks. These are structured walks through an area, under the guide of a group of local people who live there and know the area well. The itinerary can be decided before, hand, depending on the aim of the walk [57]. Field activities help to figure out living conditions, meet people and foster participation. They allow to collect more precise information about soil and environmental characteristics, problems faced by farmers through visual observation and discussion with people [14,15,24,25,43,44,47,58]. The fact that the object of discussion is often directly visualized by both sides (researchers and farmers) makes it simpler for both to reach a common understanding. Field activities, can be used at the beginning or after an initial group phase [54], to draw or to verify a map [42,45,57]. Rogé et al. [21] and Barrios et al. [10], transect walks were used to collect information on farmers' indicators. In some cases [24,25,27] they were propedeutic to the sampling design. These activities can involve a variable number of participants, but, for practical reasons groups are smaller than in the meetings. Visits can be organized for farmers to see with their own eyes management techniques that they eventually could themselves put in action [18,29] or to visit trials [59]. Exchange visits are never used alone, they are always coupled with others.

The interviews are the second most widely used instrument. A structured interview or questionnaire survey is a list of fixed questions, while a semi-structured interview uses a framework of themes to be investigated, paying more attention on what the interviewees say spontaneously, so that new ideas can come out [52]. Interviews may be individual or in groups, and can be performed during information gatherings, during transect walks and at any of the different stages and activities of the project. A tricky term is "survey", since it is commonly used as a synonym of interview, or more generally meaning a close examination on a specific issue.

During training activities the information goes from researcher/technician to farmer, with the aim to empower him/her or to put everyone in the same knowledge condition to allow a better group work [10]. In this situation the farmer is passive, so this technique is rarely used in studies based of a participatory type: better results have been observed in situations in which the farmer can perceive that he/she has a role in transferring knowledge. Sometimes the term "workshop" is incorrectly used to indicate training activities. Fairs may be important participatory tools that allow farmers from different communities to meet and share knowledge. There can be soil fairs, food fairs, fairs in which farmers explain to other farmers advantages gained by implementing innovations. Fairs are fundamental to build trust in new techniques, or in situations where traditional techniques are the answer, but these have been forgotten by some communities. They are also important for providing demonstrations for some monitoring methods and instruments, as a place for farmers and scientists to meet and build an integrated language [10] and to help farmers to develop skills to recognize and describe soil characteristics [23]. In the examined papers only two report about the organization of "soil fairs" [10,23].

To ensure a project is successful, it is recommendable to involve farmers in a testing or a technology building phase. This is done by creating a space in a public or private area to allow people to test a technique, before applying it. This builds trust in the farmers and allows a situation to be avoided in which researchers suggest solutions that the farmers will not apply, with a waste of resources [10,50].

How is the choice of these categories related to the goals of a project? To understand this, the papers have been categorized based on their main and explicit purpose. The main purposes identified can be grouped into six categories:

1.  Identification of a set of indicators. This was one of the main purposes in 10 papers. Indicators were aimed at assessing: sustainability [60], soil quality [10,16,23,43], fertility [16], acidity [24], erosion [44], structure [38], destination [45], pasture quality [46]. These are not the only papers that studied indicators, but the ones which stated that this was the main purpose. All are related to the participatory category of group activities, because they are the more effective way to identify a set of indicators.

2.  Testing the validity of local indicators—four papers indicated this as main objective: to understand how reliable local indicators are, by comparing local assessment and soil analysis results [14,16,25,26]. In all cases group activities were used to identify indicators, before testing their validity.

3.  Elicit local information—the main aim to use participatory techniques in 8 papers was to receive information on local knowledge of soil quality [47], soil quality change [25], every available information on soils characteristics [15,23] and management techniques [23,45], on natural resources management methods [44], soil fertility and fertility management [36], social issues [39]. To obtain local information, the most used participatory methods are group activities, followed by interviews.

4.  Study local perception—9 papers focused on studying how local people perceive a certain issue, such as soil acidity [24], change in soil quality [25], soil quality and fertility [31,33], new technologies for soil fertility [61], production constraints [34], land degradation/erosion [46,48] and its management [48], sustainability [58]. To study local perception, the most used participatory methods are group activities jointly with interviews.

5.  Integrate local and scientific knowledge/methods—the main objective of 2 papers [10,31] was to find a way to integrate local and scientific knowledge to make collaboration easier and more efficient. Group activities and interviews are the common participatory techniques.

6.  Improvement/facilitation/empowerment—since the participatory techniques are usually used for development projects, there are some papers (10) in which the main objective was to improve farmers conditions [21,29,59], evaluation and monitoring approaches [26,28], soil management [13,18,41,42], peasant empowerment [46].

7.  Investigation—18 papers had various objectives, concerning the investigation of different issues, such as agriculture [17,22,27,30,32,34,35,39,62], management techniques' effects [13,37,48], environmental ones [63], social ones [20,47].

There is no apparent correlation, between participatory techniques and papers' aims. This can be attributed to the fact that the papers are multidisciplinary and vary a lot from one another.

### 3.6. Involvement of Women

Women's involvement in agriculture is strongly determined by geographical location (Table 4). The average percentage of agricultural labour force represented by women is 20% in Latin America and 50% in Eastern Asia and sub-Saharan Africa, while the average share of rural households that have a female head is 25.5% in Africa [12]. In Table 4 more detailed data about every country are provided.

It is evident that woman represent a non negligible component of agricultural labour and sometimes even of household heads and farm holders. In spite of this, less than half of the studies, only 17 out of 43 [10,14,18,21,22,25,27,29,31,34,36,43,44,48,59,61,62] nearly all of which carried out in Africa, and only one in Mexico, state that both men and women were involved in the study. This may be the result of cultural challenges. Erkossa et al. (2004) [45] (Ethiopia) report that the few female-headed households in their study were too timid to show up during the workshops.

Four papers only collected sex-disaggregated data, allowing to assess the different perceptions and knowledge of men and women. Ajayi [61], Dawoe et al. [36] and Prudat et al. [31] (respectively Zambia, Ghana, Namibia) state that in their case the perceptions of farmers are not affected by gender, while Kuria et al. [43] (Rwanda) show the opposite. In particular, Kuria et al. [43] tried to understand how gender influenced indicators' choice and soil management practices and found that the gender issue may be a central one. They found significant differences in some indicators of soil quality and soil management practices. These were dictated by the gender division of tasks during the cropping cycle. This separation of duties is confirmed by the information that can be found in the FAO Gender and Land Right Database [64]. Some authors, although do not explicitly talk about involvement of women in their studies, do make observations on the different roles of men and women [18,19,25,33,58]. Even perception of soil erosion was different among genders because of the differentiated access and control over certain areas.

**Table 4.** Percentage of agricultural labour carried out by women, the percentage of households headed by women and the percentage of female land tenure in different countries.

| | Country | Female Agricultural Labour [12] | Women Headed Households [12] | Female Agricultural Holders [12] |
|---|---|---|---|---|
| Africa | Benin | 39.6% | 19.2% | \ |
| | Burkina Faso | 47.7% | 7.5% | 8.4% |
| | Ethiopia | 45.5% | 20.1% | 18.7% |
| | Ghana | 44.3% | 30.8% | \ |
| | Kenya | 48.6% | 33.8% | 5% [64] |
| | Mali | 37.7% | 11.5% | 3.1% |
| | Namibia | 44.6% | 47.4% | \ |
| | Rwanda | 57.0% | 34.0% | \ |
| | Senegal | 47.4% | 10.7% | 9.1% |
| | Tanzania | 30–40% | \ | \ |
| | Uganda | 49.5% | 29.3% [64] | 16.3% |
| | Zambia | 46.5% | 25.4% | 19.2% |
| | Zimbabwe | 53.3% | 42.6% | \ |
| Asia | Bangladesh | 51.0% | 13.2% | \ |
| | India | 32.4% | 14.9% | 10.9% |
| | Thailand | 45.0% | \ | 27.4% |
| | Vietnam | 49.1% | 22.4% | 8.8% |
| Latin America | Brazil | 24.5% | 13.7% | \ |
| | Chile | 14.2% | \ | 29.9% |
| | Colombia | 24.8% | 21.7% | \ |
| | Honduras | 20.7% | 20.2% | \ |
| | Mexico | 12.3% | \ | \ |
| | Venezuela | 6.4% | \ | \ |

Reasons for lack of involvement are also related to the fact that the roles of women may make it difficult for them to leave the house and the village, so that participatory events held in the village are more attractive for them [59] (Kenya). On the other hand, talking about actions that farmers could perform to achieve development in their villages,

Deugd et al. [42] report that actions such as developing multi-industry enterprises are typically carried out by women.

Trying to understand which are the characteristics shared by papers that have included women in their studies, no relationship can be observed with the year of publication, so there has been no apparent evolution in this aspect. On the other hand, as already mentioned, there seems to be a strong relation with the geographical area: in fact, nearly all the studies which involved women are set in Africa. There are no studies involving women in Asia and only one in Latin America. In Latin America there are the lowest values of female work labor and the Asia and Latin America countries are considered have fewer female heads of households than Africa, so that maybe the reason why involving women is more difficult and apparently less important. The only two studies set in Africa in which women have not been involved were carried out in Burkina Faso and Senegal, that record the lowest number of female heads of households among the African countries.

Considering the categories related to the main objectives of the paper, the two papers whose main objective was to integrate scientific and local knowledge involved women in the study, while the categories which had the lowest rate of involvement were: identification of indicators, investigation and collection of local information.

### 3.7. Climate Change

One of the purposes of this review was to examine the use of indicators to assess climate change, but they were not considered in the examined papers. Only three articles refer to climate change [23,46,58]. This is probably because participatory approaches are mostly used in projects related to the agricultural development or sustainable development of rural communities, and being focused on giving specific instruments and scientific knowledge to the local populations in order to enhance agricultural productivity. However, the indicators used are about soil quality, soil quality change, soil acidity, soil erosion, soil structure, soil fertility, soil workability, soil health, sustainability in general. These indicators are all closely related to climate change effects, even if the articles do not address the problem directly. When a change in soil characteristics is addressed, it is not specified if it is due to climate or management issues. In most cases researchers evaluate the present state of soil quality and allow farmers to express their perceptions with respect to the past and to monitor the future conditions.

### 3.8. Correlation among Different Characteristics of the Papers

We tried to summarize this analysis by asking ourselves: are there relationships among papers based on their characteristics? Which indicators are preferred to analyze certain problems? Are some participatory techniques more efficient to select some types of indicators? Are there some participatory techniques or indicators' categories that are more useful for certain purposes? To answer these questions, a quantitative meta-analysis has been carried out, calculating relationships among different characteristics of the papers, using the qgraph instrument with R software (Figure 4).

In Figure 4 the characteristics taken into consideration are represented with different colours, while in every cluster are reported the classes for every characteristic. In a previous attempt the indicators were represented in clusters, divided in categories based on the type of indicator (e.g., physical/chemical characteristics, etc.), but the only correlation was found inside the cluster, so these were grouped within one class. Green and red lines indicate positive and negative weights, respectively, while line width is proportional to the absolute weight and scale relative to the strongest weight in the graph. Lines with an absolute value under the minimum argument (0.5 in this case) are omitted.

The correlation between the involvement of women (W) and Africa (GL1) (16 papers) confirms the qualitative observations reported above.

The correlation between the attempt of an integration (I) and technical (TI) (17 papers) and local indicators (LI) (15 papers) confirms that, in cases where there is a will to integrate local and scientific knowledge, the use of indicators is helpful. In 14 cases, the integration

purpose (I), technical (TI) and local indicators (LI) are present together. There is an obvious negative correlation between integration (I) and the investigation purpose (P7): attention to local knowledge was not a purpose of papers whose aim was centered on investigation.

The strong correlation between the team experience 5 (TE5—engineering) and multidisciplinary journal type (JT5) is simply given by the fact that there is only one paper with an engineer in the team. The same concept explains the correlation between integration purpose (P5) and earth and planetary sciences experience (TP3), not specific geographical location (GL4) and Y1 (before or in the year 2000).

Papers published in agricultural journals (JT1), were written by larger teams (NA56) (11 papers). Studies in which experiments have been carried out by farmers (PT6) (9 papers) half the time are coupled with training sessions (PT4—4 papers), while there is no characteristic that seems to be related to climate change (CC).

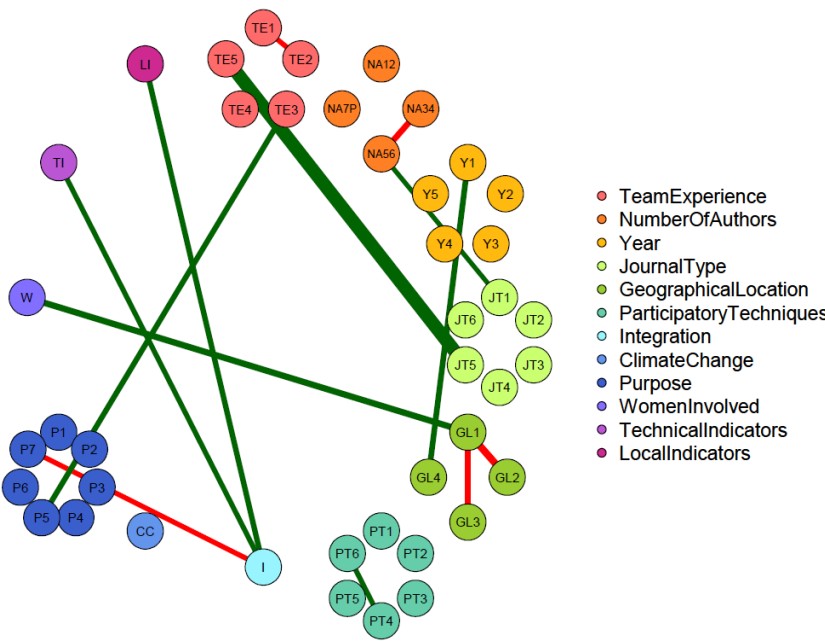

**Figure 4.** Correlation graph calculated with the qgraph tool in R software. The graph shows different levels of a factor represented with circles and grouped in clusters. The green and red lines, indicate positive and negative correlations respectively. The width of the line corresponds to the absolute weight and scale relative to the strongest weight in the graph. Lines with an absolute value of 0.5 were omitted. TE—Team Experience, TE1—Agricultural and Biological Sciences, TE2—Environmental sciences, TE3—Earth and Planetary Sciences, TE4—Social Sciences, TE5—Engineering, NA—Number of Authors, NA12—one or two authors, NA34—three or four authors, NA56—five or six authors, NA7P—seven or more authors, Y—Year, Y1—before or in the year 2000, Y2—from 2001 to 2005, Y3—from 2006 to 2010, Y4—from 2011 to 2015, Y5—from 2016 to 2020, JT—Journal Type, JT1—agriculture, JT2—environment, JT3—sustainability, JT4—geography, JT5—multidisciplinary, JT6—ethnobiology, GL—Geographical Location, GL1—Africa, GL2—Latin America, GL3—Asia, GL4—not specific, PT—Participatory Techniques, PT1—group activities, PT2—field activities, PT3—interviews, PT4—trainings, PT5—fairs, PT6—experiments, I—integration, CC—climate change, P—Purpose, P1—Identify indicators, P2—Test the validity of local indicators, P3—Elicit local information, P4—Study local perception, P5—Integrate local and scientific knowledge/methods, P6—Improvement/facilitation/empowerment, P7—Investigation, W—women, TI—technical indicators, LI—local indicators.

## 4. Conclusions

Our meta-analysis of studies carried out in developing countries by making use of participatory strategies has highlighted some methodological and structural aspects that may limit further progress in designing better projects and in developing a more scientific and structured approach to participatory techniques.

Most of the scientists that were involved so far in this type of studies were experts in agricultural, biological, and environmental sciences and despite participatory approaches were born in social sciences, none of the articles was published in social sciences journals and only two in ethnobiology journals. Less than 30% of the studies featured a collaboration with social science experts. It is probable that a more efficient use of participatory techniques and of integration of local and technical knowledge could be achieved if the collaboration with social sciences experts was more systematic.

Participatory approaches still lack a standardized nomenclature or protocol [65]. This generates confusion, as the terms used to identify the same type of activities differ from one paper to the other. Participatory activities are not only crucial for the successful implementation of cooperation projects, but in the absence of scientifically designed protocols, the approach may suffer from methodological faults or overlook the importance of factors such as the gender issue.

A notable attempt in this sense is a guide developed by Barrios et al. [10] for Latin America and Africa, which explains the techniques used to involve farmers in the selection of indicators; many papers examined are related to this or to those involving one or more of the authors [14,23,29,43], and others cite their works (the article or one of the guides) [16,25,26,28,31,33,36,38,47].

Some differences emerged in the types of indicator used by scientists and by farmers: farmers use more Pressure and Response indicators than scientists. The most used technical indicators are texture, pH, available P, organic carbon. The most used local indicators are color, texture, field location, structure, yield and productivity.

The local indicators are not standardized: there is a variety of terms and ways to assess the same characteristics; in addition, they rely only on observation. Some papers specified the assessment method, but many of them did not. This information should be included in the papers, in order to allow use by other farmers. In other words, they should be treated as the technical ones, which are univocally defined in order to permit a generalized use.

For integration the use of indicators is fundamental. Hence it is of the utmost importance to find new means to validate local indicators and translate them into a common language that can be understood by both farmers and scientists.

Women were involved only in African projects; their social role influences their knowledge and responsibilities. Given the importance of women in smallholder agriculture, it is important to take into consideration women in these studies and, whenever, possible to collect sex-disaggregated data, since the different role in society can reflect a different knowledge and perception of agricultural and environmental issues.

Finally, only three articles mentioned climate change. Given the importance for developing countries, especially for rural realities, to adapt quickly to the new risks imposed by climatic changes, it is becoming of utmost importance for a truly sustainable land use for projects to take into consideration threats posed by climate changes.

**Author Contributions:** Conceptualization, G.B., L.P. and M.D.N.; methodology, G.B., L.P. and M.D.N.; validation, G.B.; formal analysis, G.B.; investigation, G.B.; data curation, G.B.; writing—original draft preparation, G.B.; writing—review and editing, G.B., L.P. and M.D.N.; supervision, L.P. and M.D.N.; funding acquisition, L.P. and M.D.N. All authors have read and agreed to the published version of the manuscript.

**Funding:** This research was funded by Agenzia Italiana per la Cooperazione allo Sviluppo (AICS) codex AID 011.483 CUP G56C18000510001 2017 Call of the Italian Agency for Cooperation and Development for the granting of contributions to initiatives proposed by organizations of civil society and non-profit entities. 2017 financial endowment.

**Acknowledgments:** This research is part of the project "Protagonismo de las mujeres rurales para la afirmación de la soberanía alimentaria en Bolivia", in collaboration with CeVI (Centro di Volontariato Internazionale, Udine).

**Conflicts of Interest:** The authors declare no conflict of interest.

## Appendix A

**Table A1.** Correlation table with data used for Figure 4.

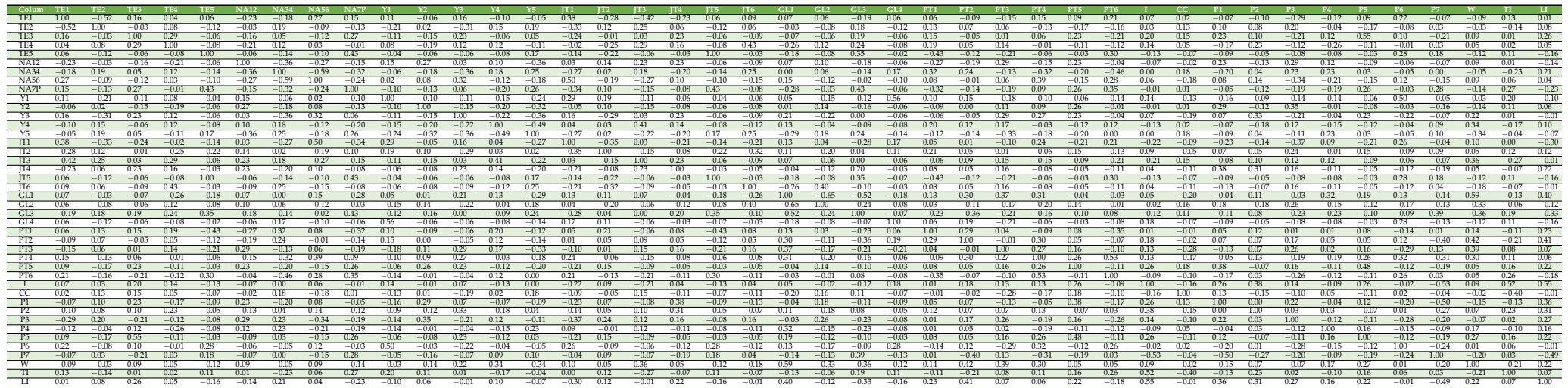

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
