# Peer review of "Integrated Use of Local and Technical Soil Quality Indicators and Participatory Techniques to Select Them. A Review of Bibliography and Analysis of Research Strategies and Outcomes"

_sustainability, doi:10.3390/su13010087_

Round 1

Reviewer 1 Report

The manuscript entitled “Integrated use of local and technical soil quality indicators and participatory techniques to select them. A review of bibliography and analysis of research strategies and outcomes” presents a review work summarizing existent knowledge in aspect of: impact of climate change on fertility of soils, assessment of quality and fertility of soils, applied approaches and techniques, as well as the involvement of women in analyzed studies.
In my opinion, the abstract should be modified and condensed to reflect the most important findings. I think it is not required to cite relevant studies in the abstract. These studies should be cited further, in the text of the manuscript.
The abbreviations should be also explained in places where they are used for the first time, for example “FAO” in line 85.
There are also some technical errors, please see line 168 and remove the sentence “Error! Reference source not found”. The same in lines 173,185, 233, 235, 255, 263, 341, 482, 485, 542, 557.
I also recommend to focus more on the originality and novelty of performed analysis. I cannot get the idea in what way the presented manuscript and the data presented would be significant to the field of agricultural study. Please justify the purposefulness and sensibility of conducting this type of analysis.

Author Response

COMMENT 1: In my opinion, the abstract should be modified and condensed to reflect the most important findings.

I think it is not required to cite relevant studies in the abstract. These studies should be cited further, in the text of the manuscript.

RESPONSE: The citations in the abstract have been removed and the abstract has been modified to highlight important findings and conclusions.

COMMENT 2: The abbreviations should be also explained in places where they are used for the first time, for example “FAO” in line 85.

RESPONSE: This has been fixed (lines 38, 86)

COMMENT 3: There are also some technical errors, please see line 168 and remove the sentence “Error! Reference source not found”. The same in lines 173,185, 233, 235, 255, 263, 341, 482, 485, 542, 557.

RESPONSE: We apologize for this problem, which is probably due to the fact that an automatized reference Word tool was used to insert the figures. Probably the use of a different version of Word caused the disappearance of the references. We removed the automatised references and inserted them manually.

COMMENT 4: I also recommend to focus more on the originality and novelty of performed analysis. I cannot get the idea in what way the presented manuscript and the data presented would be significant to the field of agricultural study. Please justify the purposefulness and sensibility of conducting this type of analysis.

RESPONSE: Following suggestions by the reviewer the abstract, the introduction and the conclusions have been revised to clarify why this review will be useful for advancement of studies on sustainability of soils in developing countries, and in particular to scientists in preparing projects aimed at helping smallholders to counteract the negative effects of climatic changes.

Reviewer 2 Report

This study aims to review how the problem of building a set of integrated indicators to assess soil quality have been faced until today, which participatory techniques are better used and, given the importance of women in agriculture.

The study potentially makes an important contribution to its field but some changes should be considered in the manuscript. The discussion should focus more on the implications of the findings in the broadest context possible and limitations of the work highlighted. Future research directions may also be mentioned. Please see the comments described below:

- The abstract should not include bibliographic references.

-Line 44. This is very general and not true “In all continents, this was the warmest period ever registered [10]”... The Planet has registered warmest periods along with its history.

-Line 58-60. It would be desirable to explain how soil could mitigate climate change.

- In a review paper more bibliographic references would be excepted in the introduction section.

- Lines 146-148- The specific dates used to carry out the literature search should be mentioned.

- Under my point of view to carry out a review study with only 12 papers it is a bit limited.

Lines 168, 173, etc. Please include the reference to the Figures.

Figure 2: Please explain in the material and method section how do you recognise  the main experience field of the authors using Scopus indications

Line 199. Under my point of view, the discussion is very descriptive. It should be more focuses on the implications of the findings in the broadest context possible and limitations of the work highlighted. Future research directions may also be mentioned.

Table 2. Please review it. For example, the texture is used as an indicator for 18 paper according to the bibliography column

Please, take care of the enumeration of the tables.

Take care with the format, for example in table 4 some categories begin with uppercase but not all.

Line 355 Where come from this enumeration? An introduction to it should be included.

Figure 6 How this correlation was carried out. It should be more widely described in material and method. (lines 544-556) It should be explained the meaning of the colours and width of the lines. The numerical data should be described as supplementary material.

The conclusion section should be summarised.

The revision of the English will improve the manuscript. Some expression sound very colloquial.

Author Response

COMMENT 1: The study potentially makes an important contribution to its field but some changes should be considered in the manuscript. The discussion should focus more on the implications of the findings in the broadest context possible and limitations of the work highlighted. Future research directions may also be mentioned. 

RESPONSE: The manuscript has been fully revised to meet the suggestions made by the reviewer. The implications of the results have been discussed to make clear why this paper can represent a useful tool in the preparation of future participated projects. Future research directions have also been properly addressed, as well as the weaknesses emerged by examining the existing literature.

COMMENT 2: The abstract should not include bibliographic references.

RESPONSE: The bibliographic references in the abstract have been removed.

COMMENT 3: -Line 44. This is very general and not true “In all continents, this was the warmest period ever registered [10]”... The Planet has registered warmest periods along with its history.

RESPONSE: The sentence has been amended by specifying that the report actually refers to post-industrial temperatures (Lines 44-45).

COMMENT 4: -Line 58-60. It would be desirable to explain how soil could mitigate climate change.

RESPONSE: The explanation can be found in lines 53-60, which have been lightly modified to be clearer.

COMMENT 5: In a review paper more bibliographic references would be excepted in the introduction section.

RESPONSE: We do not agree with the reviewer on this point. It very much depends on the topic and on how the work is organized within the manuscript.

COMMENT 6: Lines 146-148- The specific dates used to carry out the literature search should be mentioned.

RESPONSE: The date has been added in line 154

COMMENT 7: Under my point of view to carry out a review study with only 12 papers it is a bit limited.

RESPONSE: The review was carried out with 43 papers and not 12, the ones which were considered valid following the exclusion criteria. These were studied using some guide questions (lines 165-171). 12 were the papers which gave an answer to all our questions, but since the sentence about the 12 papers was misleading and not very clear, it was removed from the text.

COMMENT 7: Lines 168, 173, etc. Please include the reference to the Figures.

RESPONSE: We apologize for this problem, which is probably due to the fact that an automatized reference Word tool was used to insert the figures and the corresponding references. Probably the use of a different version of Word caused the disappearance of the references. We removed the automatised references and inserted them manually.

COMMENT 8: Figure 2: Please explain in the material and method section how do you recognise the main experience field of the authors using Scopus indications

RESPONSE: The requested information has been added in lines 192-193

COMMENT 9: Line 199. Under my point of view, the discussion is very descriptive. It should be more focuses on the implications of the findings in the broadest context possible and limitations of the work highlighted. Future research directions may also be mentioned.

RESPONSE: Implications of findings have been added at each appropriate point, as well as the shortcomings of the approaches used so far. Future research directions have also been included.

COMMENT 10: Table 2. Please review it. For example, the texture is used as an indicator for 18 paper according to the bibliography column

RESPONSE: The table has been revised and references were recounted.

COMMENT 11: Please, take care of the enumeration of the tables.

RESPONSE: We apologize for the mistakes. Enumeration of the tables has been corrected.

COMMENT 12: Take care with the format, for example in table 4 some categories begin with uppercase but not all.

RESPONSE: The table has been thoroughly revised.

COMMENT 13: Line 355 Where come from this enumeration? An introduction to it should be included.

RESPONSE: Again this was a mistake and has been corrected.

COMMENT 14: Figure 6 How this correlation was carried out. It should be more widely described in material and method.

RESPONSE: An appropriate a paragraph has been added to the materials and methods section (lines 204-210)

COMMENT 15: (lines 544-556) It should be explained the meaning of the colours and width of the lines. The numerical data should be described as supplementary material.

RESPONSE: The meaning of colours and width of lines have been described in lines 574-577, an explanation has also been added to the figure caption. The numerical data have been added in the Appendix, as suggested.

COMMENT 16: The conclusion section should be summarised.

RESPONSE: The conclusions section has been rewritten, summarizing the main outcomes of the review and to point out reasons why the results will be useful for the preparation of future participated projects on sustainable management of agricultural soils and resilience to climate change.

COMMENT 17: The revision of the English will improve the manuscript. Some expression sound very colloquial.

RESPONSE: The English language has been thoroughly revised throughout the manuscript.

Reviewer 3 Report

a deep analysis of different papers doesn't carry on to scientific results and conclusions.

the women's role appears often unrelated to the main target of the paper; it is necessary to find a common way to conclusions

it is necessary to relate the scientific analysis to people experience. and it is so difficult, as often scientific data don't macth with other data collected

Author Response

The manuscript has been revised throughout and the usefulness of the analysis should now be clearer to readers. We do not agree with the reviewer and believe that this review may represent a tool to find common more rational and holistic approaches to implement research in this field.

Round 2

Reviewer 1 Report

I am not sure if it is required to triplicate the same affiliation in lines 7-9. I think that would be better to add this affiliation once and then present e-mails of each author.

The abstract should be shown as coherent text, without division into sections and paragraphs.

I cannot see in the text which sections have been improved and how. The authors did not provide the version with changes marked. I only see the final version, so how can I refer to and check parts that I suggested to improve?

Author Response

COMMENT 1: I am not sure if it is required to triplicate the same affiliation in lines 7-9. I think that would be better to add this affiliation once and then present e-mails of each author.

RESPONSE: The affiliations have been corrected.

COMMENT 2: The abstract should be shown as coherent text, without division into sections and paragraphs.

RESPONSE: The divisions into sections have been cancelled.

COMMENT 3: I cannot see in the text which sections have been improved and how. The authors did not provide the version with changes marked. I only see the final version, so how can I refer to and check parts that I suggested to improve?

RESPONSE: We apologize for the problem, the manuscript will be uploaded with all the changes marked, the ones concerning the first revision and the two changes described above.

Reviewer 2 Report

I appreciate the revision of the manuscript. I consider that it is publishable

Author Response

We thank for the response.

Reviewer 3 Report

some little improvemets were done.